# Multi-Robot Scene Completion:
# Towards Task-Agnostic Collaborative Perception

**Yiming Li**[*]
New York University
yimingli@nyu.edu

**Juexiao Zhang**[*]
New York University
juexiao.zhang@nyu.edu

**Dekun Ma**
New York University
dm4524@nyu.edu

**Yue Wang**
Massachusetts Institute of Technology
yuewang@csail.mit.edu

**Chen Feng**[†]
New York University
cfeng@nyu.edu

**Abstract:** Collaborative perception learns how to share information among multiple robots to perceive the environment better than individually done. Past research on this has been task-specific, such as detection or segmentation. Yet this leads to different information sharing for different tasks, hindering the large-scale deployment of collaborative perception. We propose the first task-agnostic collaborative perception paradigm that learns a single collaboration module in a self-supervised manner for different downstream tasks. This is done by a novel task termed *multi-robot scene completion*, where each robot learns to effectively share information for reconstructing a complete scene viewed by all robots. Moreover, we propose a spatiotemporal autoencoder (STAR) that amortizes over time the communication cost by spatial sub-sampling and temporal mixing. Extensive experiments validate our method's effectiveness on scene completion and collaborative perception in autonomous driving scenarios. Our code is available at https://coperception.github.io/star/.

**Keywords:** Multi-Robot Perception, Scene Completion, Representation Learning

## 1 Introduction

Single robot perception has been widely studied on tasks such as object detection [1] and semantic segmentation [2]. However, it suffers from various challenges, such as occlusion and sparsity in raw observations. Collaborative perception is promising to alleviate those issues. It provides more environment observations from different perspectives by information sharing to improve perception performance and robustness. Amongst different collaboration strategies, feature-level collaboration [3, 4, 5] transmits the intermediate representations generated by deep neural networks (DNNs) of each robot. Since these intermediate features are easy to compress and can preserve contextual information of the scene, feature-level collaboration demonstrates better performance-bandwidth trade-off compared to raw-data-level and output-level collaboration [6, 7].

However, existing feature-level collaboration methods [8, 4, 3] are fully supervised by task-specific losses to learn the entire model, including a feature extractor, a collaboration module, and a decoder, as shown in Fig. 1 (a). Such a task-specific framework requires re-training the whole model for different perception tasks. Besides, existing collaborative perception requires training data recordings to be synchronized among all robots in time, which is more demanding than data collection in single-robot perception. How can we design a collaborative perception framework (1) independent from downstream tasks and (2) trainable from asynchronous datasets?

To answer this question, we propose a novel self-supervised learning task termed *multi-robot scene completion*. It enables multiple robots to collaboratively use an autoencoder to reconstruct a complete scene based on shared latent features. The completed scene could then be fed into various

---

[*]indicates equal contribution.

[†]Corresponding author

6th Conference on Robot Learning (CoRL 2022), Auckland, New Zealand.

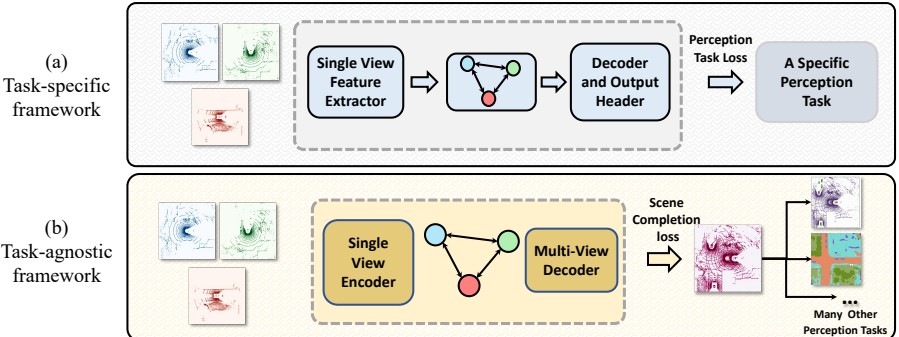

Figure 1: **Task-specific vs Task-agnostic collaboration.** Task-specific paradigm learns different models with different losses for each task. Whereas for the task-agnostic paradigm, reconstruction of the multi-robot scene is learned, which is independent of yet still usable by all downstream tasks.

downstream tasks *without additional training*, as shown in Fig. 1 (b). This allows us to decouple the collaboration training from downstream task learning. Moreover, it seamlessly supports synchronous and asynchronous training datasets with different learning objectives: complete scene reconstruction if synchronous and individual view reconstruction if asynchronous.

Yet naive autoencoders are not designed to balance scene reconstruction performance and communication volume, which is an established criterion to evaluate collaborative perception. To address this challenge, we further design a spatiotemporal autoencoder (STAR) inspired by the recent masked autoencoders (MAE) [9]. It reconstructs a scene using a spatiotemporal mixture of patch tokens: some tokens are encoded from randomly sub-sampled patches in the current frame and others are cached from the past. The sampling ensures that all patches in the mixture can jointly cover the whole spatial region while being self-disjoint. This allows each robot to only transmit the sub-sampled tokens in the current frame instead of the entire latent feature maps, leading to much lower communication bandwidth than prior works. *Our key insight behind such an amortized communication cost* is that features of many patches (e.g., static or nearly static) do not need to be shared in every frame.

In summary, our main contributions are threefold:

- We propose a brand-new task-agnostic collaborative perception framework based on multi-robot scene completion, decoupling the collaboration learning from downstream tasks.
- We propose asynchronous training and synchronous inference with a shared autoencoder to solve the proposed task, eliminating the need for synchronous data for collaboration learning.
- We develop a novel spatiotemporal autoencoder (STAR) that reconstructs scenes based on temporally mixed information. It amortizes the spatial communication volume over time to improve the performance-bandwidth trade-off.
- We conduct extensive experiments to verify our method's effectiveness for scene completion and downstream perception in autonomous driving scenarios.

## 2   Related Works

**Collaborative perception.** Collaborative perception has been proposed to improve individual perception's flexibility, resilience, and efficiency [10, 11, 12]. With recent advances in deep learning, researchers have developed feature-level collaborative perception in which intermediate representations produced by deep neural networks (DNNs) from multiple viewpoints are propagated in a team of robots, *e.g.*, a swarm of drones [8, 3] or a group of vehicles [4, 6]. Existing works commonly consider a specific downstream task and use the corresponding loss function to learn a collaboration module such as a graph neural network (GNN) [4, 3], a Transformer [13, 14], and a convolutional neural network [5, 15]. Several downstream tasks have been investigated in collaborative scenarios, such as object detection [4], semantic segmentation [8], and depth estimation [3]. Unlike the existing task-specific collaborative perception sharing task-dependent representations, we define task-agnostic [3] collaborative perception as the feature-level collaborative perception sharing task-independent representations amongst multiple robots.

---

[3]Herein "task" denotes the downstream perception task such as object detection and semantic segmentation.

**Scene completion.** Autonomous navigation [16] requires robots to understand the geometry and semantics of 3D scenes. However, vision sensors only capture partial observations because of a limited field of view and sparse sensing, leading to an incomplete spatial representation. Therefore, scene completion (SC) has been proposed to infer the complete 3D scene geometry given sparse 2D/3D observations [17, 18, 19]. Following scene completion, semantic scene completion (SSC) has been introduced to jointly estimate both geometry and semantic information based on partial observation [2, 20, 21, 22]. On the one hand, single robot scene completion can rely on prior semantic knowledge to complete the partially-observed objects. On the other hand, it is unrealistic to see through full occlusions. Unlike single robot scene completion depending on prior knowledge, the multi-robot scene completion task utilizes information shared by teammates for scene completion.

**Self-supervised representation learning.** Self-supervised representation learning (SSRL) aims to provide powerful features without the need for massive annotated datasets [23]. SSRL is generally composed of: (1) *task-agnostic* pre-training via carefully-designed self-supervised pretext tasks such as contrastive learning [24, 25] or autoencoding [26, 27, 9], and (2) *task-specific* adaptation to fine-tune the pre-trained model on the downstream tasks such as object detection or image classification. Masked autoencoder (MAE) achieves great performance with a simple reconstruction objective [9]. It employs an asymmetric architecture with a large encoder that only processes unmasked patches and a lightweight decoder that reconstructs the masked patches from the latent representation. Recent works extend MAE into multimodal representation learning [28, 29], video [30, 31], and 2D image completion [32]. In this work, we employ similar autoencoding to learn the shared representations and enable quick adaptation to the downstream perception: the reconstructions could be seamlessly utilized by off-the-shelf individual perception model trained on single-view data without any fine-tuning, bridging the gap between collaborative perception and individual perception.

## 3 Multi-Robot Scene Completion: Motivation, Formulation, and Evaluation

**Motivation.** Even though single-robot data already requires arduous annotations like 3D bounding boxes and pixel-wise semantic labeling, multi-robot data even demands multiple times as much work. To relax the task-dependent supervision for collaboration learning, we propose multi-robot scene completion to enable task-agnostic collaborative perception. It can utilize self-supervision to learn shared representations instead of expensive task-dependent supervision. We will introduce its overall workflow, training objective, and evaluation metrics hereafter.

**Problem setup.** We consider $N$ robots in the same geographical location simultaneously perceiving the 3D environment, such as a fleet of autonomous vehicles located at a crossroad. These robots communicate with each other about their observations to better understand the surrounding environment. Each robot indexed by $i$ is equipped with a 3D sensor such as a LiDAR to generate a binary occupancy grid map $\mathbf{M}_i \in \{0, 1\}^{H \times W \times C}$ defined in its local coordinate, where $H$, $W$, and $C$ respectively denote the length, width, and height resolution.

**Feature extraction.** We employ intermediate collaboration with better performance-bandwidth trade-off [5]. Each robot encodes its observation into a feature map denoted by $\mathbf{F}_i = \boldsymbol{\Theta}(\mathbf{M}_i)$, where $\boldsymbol{\Theta}$ is a feature extractor. Now $\mathbf{F}_i \in \mathbb{R}^{\bar{H} \times \bar{W} \times \bar{C}}$ has lower spatial resolution $\bar{H} \times \bar{W}$, while keeping a higher feature dimension $\bar{C}$ compared to the original map $\mathbf{M}_i$. Then, each robot will broadcast $\mathbf{F}_i$ to its peers as well as its pose $\boldsymbol{\xi}_i \in \mathfrak{se}(3)$ defined in the global coordinate.

**Feature decoding.** The robot $i$ receives the messages from the neighboring robots $\{\mathbf{F}_j, \boldsymbol{\xi}_j\}_{j \neq i}$, and then uses a decoder $\boldsymbol{\Phi}$ and a pose-aware aggregator $\mathcal{A}$ for fusion, and output a completed occupancy grid map $\hat{\mathbf{Y}}_i = \boldsymbol{\Phi}(\mathcal{A}(\mathbf{F}_i, \boldsymbol{\xi}_i, \{\mathbf{F}_j, \boldsymbol{\xi}_j\}_{j \neq i}))$, where $\hat{\mathbf{Y}}_i$ has the same dimension and describe the same spatial range as $\mathbf{M}_i$ yet is a more comprehensive spatial representation for the scene. The pose-aware aggregator $\mathcal{A}$ transforms the feature maps of robot $j(j \neq i)$ into the coordinate of the target robot $i$, then sums all the coordinate-synchronized feature maps:

$$\mathcal{A}(\mathbf{F}_i, \boldsymbol{\xi}_i, \{\mathbf{F}_j, \boldsymbol{\xi}_j\}_{j \neq i}) = \mathbf{F}_i + \sum_{j=1}^{N-1} \Gamma_{j \to i}(\mathbf{F}_j), \tag{1}$$

where $\Gamma_{j \to i} \in SE(3)$ is the transformation from robot $j$'s coordinate to $i$'s, obtained by the exponential map of poses $\boldsymbol{\xi}_j$ and $\boldsymbol{\xi}_i$. Bi-linear interpolation is used to transform a discrete map, and the positions out of the spatial range $\bar{H} \times \bar{W}$ after transformation are padded with zero.

**Training loss.** We treat the scene completion task as a binary classification problem and use cross-entropy loss to train a neural network composed of $\Theta$ and $\Phi$. Specifically, the ground-truth $\mathbf{Y}_i \in \{0,1\}^{H \times W \times C}$ defined in the coordinate of robot $i$ represents a multi-view occupancy voxel grid with two classes, *i.e.*, free and occupied. Therefore, the loss can be computed by:

$$\mathcal{L} = -\sum_{i=0}^{N-1} \sum_{k=0}^{L-1} \sum_{c=0}^{1} y_{i,k,c} log\left(\frac{e^{\hat{y}_{i,k,c}}}{\sum_c e^{\hat{y}_{i,k,c}}}\right), \tag{2}$$

where $i$ is the robot index, $k$ is the voxel index, $L$ is the total number of the voxel ($L = H \times W \times C$), $c$ is the number of class (2 in our case), $\hat{y}_{i,k,c}$ is the predicted logits for the $k$-th voxel belonging to class $c$, $y_{i,k,c}$ is the $k$-th element of $\mathbf{Y}_i$ and is a one-hot vector ($y_{i,k,c} = 1$ if voxel $k$ of robot $i$ belongs to class $c$). Here we show the training objective using synchronous multi-robot data created by aggregating multi-robot observations based on robots' poses similar to Eq. (1): $\mathbf{Y}_i = \mathcal{A}(\mathbf{M}_i, \boldsymbol{\xi}_i, \{\mathbf{M}_j, \boldsymbol{\xi}_j\}_{j \neq i}) \, (j < M)$, where $M$ is the number of robots for the ground truth generation. Note that $\Theta$ and $\Phi$ can also be trained by individual view reconstruction on asynchronous data as shown in Eq. (3), while being deployed on synchronous data for inference as discussed in Section 4. Hallucinating the invisible scene is possible if the ground-truth $\mathbf{Y}_i$ is generated with more robots than those involved in the training phase, *i.e.*, $M > N$. Yet only synchronous training can achieve this because the ground truth requires aggregation of different viewpoints obtained synchronously.

**Evaluation metrics.** We follow the evaluation protocol in single-robot scene completion [19, 33], which uses the voxel-level intersection over union (IoU) between predicted voxel labels $\hat{\mathbf{Y}}_i$ and ground truth labels $\mathbf{Y}_i$ for each robot. Note that only non-empty voxels are evaluated.

## 4 STAR: Spatiotemporal Autoencoder

In addition to the multi-robot scene completion task, we also propose a novel architecture called **S**aptio**t**emporal **a**utoencode**r** (STAR) to tackle this problem. We will present our key design motivation, detailed modules, training, and inference procedures.

### 4.1 Design rationale

**Partially broadcasting.** Inspired by the idea of "masking" in MAE [9], we employ a similar asymmetric design as MAE yet with different purposes: MAE is to design a nontrivial self-supervisory task for pre-training via randomly masking, while the goal of STAR is to reduce the communication volume in multi-robot systems via partial broadcasting. More specifically, STAR deploys an encoder at the *sender robot* to map the entire observation to an intermediate feature representation that is selectively transmitted to lower the bandwidth. Meanwhile, STAR deploys a decoder at the *receiver robot* that reconstructs the original observation from the received partial representation.

**Spatiotemporal amortization.** Simply applying random masking to a complete observation is not a good idea. Unlike MAE mainly for object-level recognition, we aim at large-scale dynamic scene modeling. Once objects are completely masked during encoding, the decoder cannot hallucinate the corresponding objects without such knowledge. Therefore, we propose to exploit *historical tokens* to replace *mask tokens* during decoding. In this way, we can amortize the communication cost over the temporal domain by spatial sub-sampling and temporal mixing. Specifically, at the sender's side, a part of the spatial patches are sampled from the entire observation in each timestamp. Only the features of these subsampled spatial patches are communicated with the teammates. Upon receiving the transmitted patch features, the receiver's decoder combines these features with the historical patch features and reconstructs an entire observation. Since the patches are subsampled in a spatially complementary manner, the temporally mixed patches jointly cover the whole spatial region.

**Synchronization-free training.** Traditional collaborative perception approaches consider a synchronization training strategy that requires synchronous (potentially with a slight temporal latency) multi-robot recordings to train a feature-space collaboration strategy with task-specific loss functions [4, 5]. In this work, we try to relax the requirement that multiple robots simultaneously capture perception data by using single-view observations as the supervision during the training phase. We will describe this in detail in section 4.3.

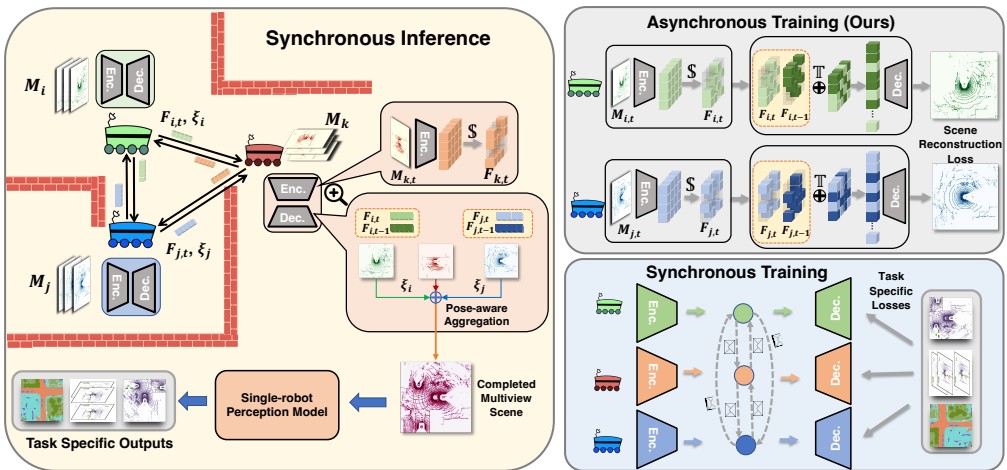

Figure 2: **Asynchronous training and synchronous inference.** In the top right, asynchronous training does not require communication between robots. In contrast, in the bottom right, **synchronous training** requires communication and optimization w.r.t. each specific task loss. The **synchronous inference** is illustrated on the left. The *sender* transmit encoded representations to the *receiver*. The *receiver* uses a mixture of spatiotemporal tokens to complete the scene observation. $\mathbb{S}$: spatial sub-sampling. $\mathbb{T}$: temporal mixing.

## 4.2 Architecture

We consider a set of robots deploying the same neural network following [4, 5]. Each robot serves as both message sender and receiver during collaboration and is equipped with an encoder for observation abstraction and a decoder for view reconstruction.

**STAR encoder.** Different from MAE [9], the STAR encoder uses a vision transformer (ViT) [34] backbone, which operates on all patches yet only sends out a subset (*spatial sub-sampling*). Specifically, the entire grid map for robot $i$ at time $t$ denoted by $\mathbf{M}_{i,t}$ is divided into multiple patches, and each patch is encoded with a linear projection with additional positional embedding, and then processed using a series of Transformer blocks to generate the final message $\mathbf{F}_{i,t}$. Note that we adopt a complementary transmission strategy in the temporal domain regarding the patch index (*i.e.*, the observed spatial locations) to avoid the loss of information for the dynamic scenes.

**STAR decoder.** Different from MAE which uses mask tokens to replace the missed patch embeddings, robot $i$ as a receiver aggregates the historic tokens $\mathbf{F}_{j,t-1}$ and the current tokens $\mathbf{F}_{j,t}$ from robot $j$ (*temporal mixing*), which approximately form a complete observation towards the entire spatial range. Temporal embeddings are added to the tokens from the respective timestamps to enhance the temporal awareness before feeding them into Transformer blocks. Note that here we use two-timestamp $t$ and $t-1$ as an example for a simple explanation. The STAR decoder is also able to process more historical timestamps. After decoding all robots' views $\{\hat{\mathbf{M}}_{j,t}\}_{j \neq i}$, the ultimate prediction of the complete view is computed by coordinate synchronization: $\hat{\mathbf{Y}}_i = \mathcal{A}(\mathbf{M}_{i,t}, \boldsymbol{\xi}_i, \{\hat{\mathbf{M}}_{j,t}, \boldsymbol{\xi}_j\}_{j \neq i})$, and its calculation process is similar to Eq. (1).

## 4.3 Training and inference

**Asynchronous training.** The model is trained with single view ground-truth $\mathbf{M}_i$, and adopt cross-entropy loss during training:

$$\mathcal{L} = -\sum_{i=0}^{N-1} \sum_{k=0}^{L-1} \sum_{c=0}^{1} m_{i,k,c} log(\frac{e^{\hat{m}_{i,k,c}}}{\sum_c e^{\hat{m}_{i,k,c}}}), \tag{3}$$

where $i$ is the robot index, $k$ is the voxel index, $L$ is the total number of the voxels, $c = 2$ is number of class, $m_{i,k,c}$ denotes the $k$-th element of $\mathbf{M}_i$ and is a one-hot vector same as $y_{i,k,c}$ in Eq. 2, $\hat{m}_{i,k,c}$ is the prediction for the $k$-th voxel belonging to class $c$. Note that the training loss is calculated voxel-wise with respect to the self-supervision signal from each robot's single-view observation. This design decouples the training phase from communication with other robots: the model on each robot does not require synchronous observations from neighbor robots in the training

phase, making the training asynchronous (asynchronous training in Fig. 2). This is greatly different from the training framework in previous collaborative perception works such as [5] (synchronous training in Fig. 2). Our training framework can relax the need for the carefully-collected and hard-to-annotate multi-robot dataset and can exploit a large amount of single-robot data to learn powerful as well as compact feature representations.

**Synchronous inference.** During inference, each robot is equipped with the same model. The sender robots' encoders will encode and broadcast a subset of their current timestamp's observation. Then, the decoders on the receiver side will leverage the transmitted intermediate representation along with the pose information to reconstruct the corresponding view, optionally with historical features as described above. Then, the receivers use corresponding pose information to aggregate the single observations into a multi-view completed scene. We illustrate the pipeline on the left side of Fig. 2.

## 5 Experimental Results

### 5.1 Experimental setup

**Dataset.** We conduct experiments on the V2X-Sim Dataset [15], a large-scale dataset that simulates urban multi-vehicle driving scenes with CARLA [35]. We use 80 scenes for training and ten scenes for testing. The dataset is sampled at 5 Hz. We pre-process the voxels grids with range $[-32m, 32m]$ in the x and y-axis and $[-3m, 2m]$ in the z-axis. Finally, we can get the voxel grids with a spatial resolution of $256 \times 256 \times 13$.

**Baselines.** The lower-bound refers to a single-robot perception model trained and tested using only individual observations. The **task-specific models** all optimize the collaboration based on the specific perception head. When2com [8] uses the attention mechanism to fuse the collaborators' information. Who2com [36] employs a handshake mechanism. The V2VNet [4] trains a graph neural network to propagate the agent's information. DiscoNet [5] selectively fuse messages from the informative regions. For **task-agnostic models**, we use a modified FaFNet [37] backbone as the CNN baseline and substitute the detection head with a classification head that outputs the logits for binary classification. VQ-VAE [26] learns a variational autoencoder to reconstruct the scene and employs a vector quantization technique to reduce communication costs.

**Implementation details.** A 6-block ViT encoder with hidden dimension 384 is used for the STAR encoder. Then an MLP is used to compress the intermediate representations to 32 dimensions and feed them to the decoder, where they are projected back to 256 dimensions and sent to a 4-layer transformer decoder. An FaFNet [37] is used for single-robot object detection. A UNet [38] serves the same purpose for the semantic segmentation task. Note that all the perception models take the three-dimensional voxel grids as input and output results in bird's eye view (BEV), such as bounding boxes and semantic labels. Our models are all trained on single-view data.

**Evaluation metrics.** For the scene completion task, we measure the completion quality using the intersection-over-union (IoU) at three different scales by down-sampling the voxels accordingly. For the perception task, we report the average precision (AP) at thresholds 0.5 and 0.7 for vehicle detection, IoU for the vehicle category, and the overall mIoU for semantic segmentation.

### 5.2 Quantitative results on scene completion

We present the quantitative results of the multi-robot scene completion task in Table 1, measured by IoU at different scales and the corresponding communication bandwidth.

**Spatial resolution.** Among the three tested resolutions, we can see that, in general, a higher spatial resolution leads to a better completion quality: the spatial resolution $32 \times 32$ which has a patch size of 8 achieves the best performance.

**Timestamps.** Our method allows the multi-robot system to amortize the spatial communication bandwidth over the temporal domain. From Table 1, we can see that from timestamps 1 to 4, the performance only varies slightly while largely reducing the bandwidth.

**Bandwidth**. Bandwidth is calculated to reflect the required data volume for communication per second. A trade-off between performance and communication is clear. The CNN baseline requires much higher bandwidth because multiple feature maps are transmitted during communication due

| Timestamp | IoU scale 1:1 | | | IoU scale 1:2 | | | IoU scale 1:4 | | | Communication Bandwidth | | |
|---|---|---|---|---|---|---|---|---|---|---|---|---|
| | 32x32 | 16x16 | 8x8 | 32x32 | 16x16 | 8x8 | 32x32 | 16x16 | 8x8 | 32x32 | 16x16 | 8x8 |
| STAR TS1 | **55.13** | 53.11 | 50.79 | **77.40** | 72.16 | 66.55 | **83.28** | 79.30 | 73.33 | 1.3MB/s | 320.0KB/s | 80.0KB/s |
| STAR TS2 | **54.93** | 52.07 | 50.40 | **75.71** | 69.63 | 64.86 | **82.51** | 76.24 | 70.81 | 640.0KB/s | 160.0KB/s | 40.0KB/s |
| STAR TS3 | **53.35** | 51.56 | 50.39 | **72.52** | 68.19 | 64.75 | **79.20** | 74.51 | 70.55 | 427.0KB/s | 106.7KB/s | 26.7KB/s |
| STAR TS4 | 53.65 | 51.64 | 49.69 | 72.98 | 68.15 | 63.39 | 79.73 | 74.36 | 68.98 | **320.0KB/s** | **80.0 KB/s** | **20.0KB/s** |
| CNN backbone | | 55.37 | | | 77.17 | | | 83.51 | | | 155.0MB/s | |

Table 1: **Quantitative results on scene completion.** Results across different spatial resolutions and timestamps are presented. Note TSX means fusing temporal information across X TimeStamps.

| Timestamp | All | Partial |
|---|---|---|
| 1 | **65.19** | - |
| 2 | **64.68** | 61.36 |
| 3 | **64.53** | 63.77 |

(a) **Patches to encode.** All: encodes all. Partial: only encodes those patches being transmitted.

| Timestamp | Multi | Single |
|---|---|---|
| 1 | **65.19** | - |
| 2 | **64.68** | 52.07 |
| 3 | **64.53** | 51.97 |

(b) **Timestamps to decode.** For timestamp 1, decoding a single timestamp is equivalent to multi.

| Timestamp | Temporal Emb. w/ | w/o |
|---|---|---|
| 2 | **64.68** | 64.29 |
| 3 | **64.53** | 61.83 |

(c) **Temporal embedding.** W/ means temporal embeddings are added. W/o means not.

| Strategy | Timestamp 2(50%) | 3(66%) | 4(75%) |
|---|---|---|---|
| random | 50.88 | 52.36 | 52.14 |
| compl. | **64.45** | **64.20** | **63.27** |

(d) **Masking strategy.** The ratio of random masking is set equivalent to complementary masking.

Table 2: **Ablation studies.** The performance is reported in IoU 1:2 for the spatial resolution 16x16. The observations under other settings are consistent.

to the skip connections in the model. STAR requires much lower bandwidth. A finer-grained spatial resolution with better performance requires a higher bandwidth.

## 5.3 Ablation studies on scene completion

We conduct several ablation studies to investigate the effectiveness of the key components in our method. Results are presented in Table 2 and are discussed in detail below.

**Patches to encode.** As shown in Table 2a, only encoding the patches that will be transmitted can result in a minor drop in performance. Yet it can reduce some computations for computation-restricted robotic systems.

**Timestamps to decode.** We investigate the effect of whether the decoder incorporates previous timestamps or just the current single timestamp combined with learnable mask tokens. Results in Table 2b indicate that historical information is essential.

**Temporal embedding.** In the STAR decoder, we add temporal embedding to the patches of different timestamps, similar to the approach in [30, 31]. The ablation study in Table 2c shows that adding temporal embedding is beneficial.

**Masking strategy.** We compared our complementary masking strategy with the random masking strategy proposed in MAE [9] in Table 2d. Results show that switching from complementary to random masking leads to a degradation in the completion performance.

## 5.4 Quantitative results on downstream perception

We directly feed the single-view to the single-robot perception model termed *lower-bound* without any fine-tuning, and the results are shown in Table 3. Our best STAR method improves the lower-bound by 25.9% and 22.8% in object detection (AP@IoU=0.7) and semantic segmentation (IoU of vehicle) respectively. Achieved by simply combining the completion model with off-the-shelf single-robot perception models, these improvements are promising because our framework: (1) has no knowledge about downstream tasks (*task-agnostic*); (2) does not require synchronous data in the training phase (*synchronization-free*); (3) is learned without manual annotations (*self-supervised*). We also investigated the single-robot perception model directly taking ground truth multi-view measurements without additional training, termed *upper-bound*, and find that it can achieve nearly com-

| Paradigm | Method | Detection | | Semantic Segmentation | |
|---|---|---|---|---|---|
| | | AP@IoU=0.5 | AP@IoU=0.7 | Vehicle | mIoU |
| Single-robot perception | Lower-bound | **49.90** | **44.21** | **45.93** | **36.64** |
| Task-specific multi-robot perception | When2com [8] | 44.02 | 39.89 | 47.87 | 34.49 |
| | Who2com [36] | 44.02 | 39.89 | 47.84 | 34.49 |
| | V2VNet [4] | 68.35 | 62.83 | **58.35** | 41.17 |
| | DiscoNet [5] | **69.03** | **63.44** | 55.84 | **41.34** |
| Task-agnostic multi-robot perception | STAR TS1 | **62.84** | **57.22** | **56.41** | **39.09** |
| | STAR TS2 | **61.48** | **55.75** | **56.13** | **38.97** |
| | VQ-VAE | 60.27 | 54.08 | 55.40 | 38.48 |
| | CNN baseline | 59.85 | 54.05 | 54.61 | 38.32 |
| | Upper-bound | **65.09** | **60.26** | **60.34** | **40.45** |

Table 3: **Quantitative results on downstream tasks.** The task-specific methods achieve excellent results via elaborate supervised learning with synchronous multi-robot recordings. The task-agnostic methods use single-robot perception models with reconstructed observations.

parable performance with DiscoNet [5] and V2VNet [4], both trained with full supervision using synchronous data for specific tasks. This demonstrates the potential of our proposed task: when the completions approach the ground truth scenes, it can perform similarly to the upper bound on many downstream tasks. Moreover, using a stronger single-robot perception model can further enhance the final performance.

### 5.5 Qualitative results on scene completion and downstream perception

We present a few qualitative results on the scene completion and the downstream tasks in Figure 3. Due to the limited space, more visualizations can be found in the appendix.

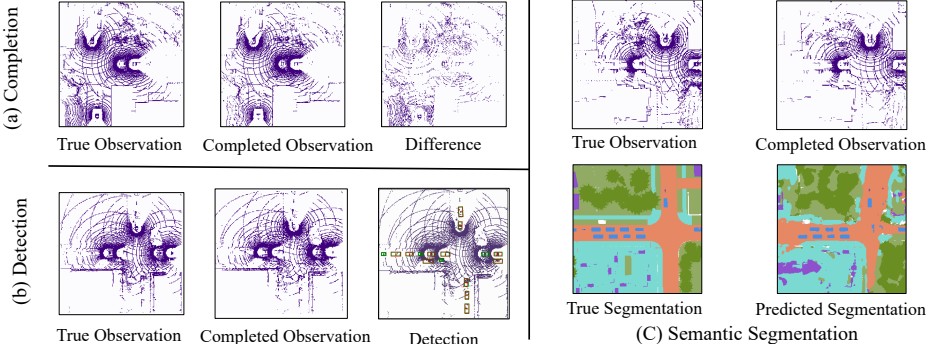

Figure 3: Qualitative results. (a), (b) and (c) each presents a qualitative example of the scene completion, detection, and segmentation tasks respectively. Refer to the appendix for more visualizations.

## 6 Limitation

There is still a performance gap between our method and the upper bound on the downstream perception due to the non-perfect scene completion. We believe when trained with more single-robot recordings, our method is able to achieve comparable performance to task-specific approaches while maintaining excellent flexibility. Currently the spatial tokens are sub-sampled randomly at individual timestamp, and we believe this could be improved in the future. We also inherit the common limitation in most existing collaborative perception works: all experiments are on simulated datasets due to the lack of public real-world datasets. We further ignore the influence of pose noises, although previous works [5] already revealed reasonable robustness.

## 7 Conclusion

We propose the first task-agnostic collaborative perception paradigm, where a single collaboration module is learned and can be transferred to different downstream tasks. Our key observation is that we can move communication between robots to the temporal domain, which achieves an excellent performance-bandwidth trade-off. Also, our self-supervised learning method sheds new light on collaborative perception that reduces the importance of human annotations.

## Acknowledgments

We thank the anonymous reviewers for their valuable comments in revising this paper. This work was supported by the NSF CPS Program under Grant CMMI-1932187 and CNS-2121391.

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
