# OpenReview forum: "Multi-Robot Scene Completion: Towards Task-Agnostic Collaborative Perception"
_robot-learning.org/CoRL/2022/Conference — CoRL 2022 Poster_

### Official Review · Reviewer_vXCo · 2022-07-15

**Originality:** Fair
**Technical Quality:** Fair
**Clarity Of Presentation:** Fair
**Impact:** 2

**Recommendation:**

Weak Reject: I recommend rejecting the paper, but will not argue for my recommendation if the majority of other reviewers have a different opinion.

**Summary:**

The paper proposes a task-agnostic collaborative perception paradigm that can support different downstream tasks through learning collaborative scene completion. To amortize the communication cost, a spatial-temporal-aware autoencoder (STAR) is proposed. The approach can be summarized as follows:
- A grid map observation from a robot is divided into patches, each patch is encoded using the STAR encoder and then a subset of patches are selected to be broadcasted along with the robot's global pose to all other robots.
- The receiving robots aggregate the information from all other robots in addition to previously received information using the STAR decoder to form the full observation
- The model is trained asynchronously on each robot.


**Issues:**

See above

**Quality Of The Limitations Section:**

Limitations are addressed clearly

**Reviewer Expertise:**

3: The reviewer is fairly confident that the evaluation is correct

**Robotics Focus:**

Relevant but unlikely to deploy to hardware in near future

**Strengths And Weaknesses:**

The paper shows a lot of promise regarding aggregating information from multiple sources that is beneficial for multiple downstream tasks. The work is also well motivated and structured, however a number of technical clarifications and experiments are missing:
1. How does the pose-aware aggregator work?
2. It was not very clear if the inference is happening synchronously or asynchronously
3. What is spatial sub-sampling and temporal mixing?
4. Given that not all information is transmitted and at the decoder side, historic tokens are aggregated with the current ones to predict the scene, how is the decision made over which patches should be transmitted? As mentioned in the paper, entirely masking a dynamic object means it cannot be properly reconstructed at decoding time. Similarly, given the dynamic nature of the scene one can imagine that the historic tokens can be no longer relevant or contain outdated scene information if the selection strategy is faulty.
5. How does the CNN baseline differ from STAR?
6. In the results, it's a bit misleading to say that STAR achieves roughly the same performance as V2V and DiscoNet when the difference in AP is around 10%

**Summary Of Recommendation:**

The idea of the paper is quite intriguing, however the current draft is lacking quite a bit of detail that is pertinent to understanding the technical contribution of the work.

---

> ### Author Response · Authors · 2022-08-26
> **Thank you sincerely for the constructive comments!**
>
> **1. How does the pose-aware aggregator work?**
>
>  Thanks for pointing this out. We add more details in our paper, see line 126-131.
>
> ---
>
> **2. It was not very clear if the inference is happening synchronously or asynchronously**
>
> Both asynchronous and synchronous training employ synchronous inference: during inference, multiple robots use the same encoder to independently convert each individual view into a feature representation, and then share the feature with teammates for the same decoder to complete the scene in parallel. We have clarified this in the caption of Figure 2, and in Section 4.3.
>
> ---
>
> **3. What is spatial sub-sampling and temporal mixing?**
>
> Thank you for the question. We add them in Section 4.2.
>
> *Spatial subsampling* is that in each timestamp, we subsample a subset of the BEV patches out of the entire BEV feature map. Only the features of these subsampled patches are transmitted  to teammates.
>
> *Temporal mixing* is that upon receiving the transmitted patch features, the decoder combines these features with the historical patch features received from previous timestamps and reconstructs an entire observation. Since the patches are subsampled in a spatially complementary manner, the temporally mixed patches jointly cover the entire spatial region (see our response to Reviewer MFnL (b) “[A, B; C, D]”).
>
> ---
>
> **4. Given that not all information is transmitted and at the decoder side, historic tokens are aggregated with the current ones to predict the scene, how is the decision made over which patches should be transmitted? As mentioned in the paper, entirely masking a dynamic object means it cannot be properly reconstructed at decoding time. Similarly, given the dynamic nature of the scene one can imagine that the historic tokens can be no longer relevant or contain outdated scene information if the selection strategy is faulty.**
>
> Thank you for the question. The patches are masked and transmitted in a **complementary** way. In particular, take the setting of 2 timestamps (t=0 and t=1) as an example, at t=0 we will randomly select and transmit 50% of all the patches. Then at t=1, the other 50% of the patches that are complementary in terms of spatial locations are transmitted (see our response to Reviewer MFnL  (b) “[A, B; C, D]”). In this way we make sure that no spatial location is completely overlooked during the temporal amortization. Additionally a spatial embedding and a temporal embedding are combined to each patch similar to the approach in [1-3]. Ablation studies in Table 2 shows the effectiveness of the complementary masking strategy and the temporal embedding technique.
>
> Indeed as you pointed out, this masked communication mechanism leads to a trade-off between the perception performance (e.g., missing critical spatial locations with dynamic objects) and the communication volume (i.e., no need to repeatedly transmit features of static spatial locations). But our experiment results show the benefits of being able to flexibly control such balances without compromising the perception performance significantly even with our current random selection strategy (thanks to the high redundancy/correlation within sequential sensor observations). In our future work, we will further improve our method with a more strategic patch selection.
>
>
> [1] He, Kaiming, et al. "Masked autoencoders are scalable vision learners." Proceedings of the IEEE/CVF Conference on Computer Vision and Pattern Recognition. 2022.
>
> [2] Feichtenhofer, Christoph, et al. "Masked Autoencoders As Spatiotemporal Learners." arXiv preprint arXiv:2205.09113 (2022).
>
> [3] Tong, Zhan, et al. "Videomae: Masked autoencoders are data-efficient learners for self-supervised video pre-training." arXiv preprint arXiv:2203.12602 (2022).
>
> ---
>
> **5. How does the CNN baseline differ from STAR?**
>
> Thanks for the question. As we mentioned in the manuscript, a popular CNN network, the FaFNet [1], is used as the CNN baseline as it provides strong results on both detection and segmentation tasks. The STAR differs from the FaFNet in the following ways:
>
> (a) It uses the vision transformer architecture as the backbone for scene encoding.
>
> (b) It allows for encoding and transmitting partial representations across multiple timestamps. So that the communication is amortized in both spatial and temporal measures. The CNN baseline on the other hand has to communicate the multi-scale feature maps in every timestamp, thereby consuming more bandwidth.
>
> [1] Luo, Wenjie, Bin Yang, and Raquel Urtasun. "Fast and furious: Real time end-to-end 3d detection, tracking and motion forecasting with a single convolutional net." Proceedings of the IEEE conference on Computer Vision and Pattern Recognition. 2018.

---

> > ### Author Response · Authors · 2022-08-26
> > **Thank you sincerely for the constructive comments!**
> >
> > **6. In the results, it's a bit misleading to say that STAR achieves roughly the same performance as V2V and DiscoNet when the difference in AP is around 10%.**
> >
> > Thank you for the question. After the initial submission we have explored more choices of the specific widths and depths of the transformer architecture and obtained better results. The table below contains the updated results where you can see that the current results outperform the CNN baseline and reach 60+ in AP for the detection task.  We have also updated the results in Table 1 and 3 in the revised manuscript. We agree that there is still a performance gap between the task-specific methods and our task-agnostic method but we would also like to note that **it is not fair to strictly compare these two approaches since the former is directly using supervision from the downstream tasks.**
> >
> > Additionally, we did not claim that the performance was “roughly the same” as V2V and DiscoNet. What we said in the manuscript is that the performance of the “upper-bound”, which is a **single-robot perception model** fed with the **original multi-robot measurements** combined from all robots **without additional training**, is nearly comparable to that of the V2V and DiscoNet. This is used to demonstrate the potential/upper-limit of our method: when the completion quality is close to the ground truth, the performance on the downstream tasks can get close to the upper-bound.
> > | Method       | Det AP@IoU=0.5 | Det AP@IoU=0.7 | Seg Vehicle | Seg mIoU  |
> > |--------------|----------------|----------------|-------------|-----------|
> > | Lower-bound  | 49.90          | 44.21          | 45.93       | 36.64     |
> > | STAR TS1     | **62.84**      | **57.22**      | **56.41**   | **39.09** |
> > | STAR TS2     | **61.48**      | **55.75**      | **56.13**   | **38.97** |
> > | VQ-VAE       | 60.27          | 54.08          | 55.40       | 38.48     |
> > | CNN baseline | 59.85          | 54.05          | 54.61       | 38.32     |
> > | Upper-bound  | **65.09**      | **60.26**      | **60.34**   | **40.45** |

---

### Official Review · Reviewer_uiAy · 2022-07-30

**Originality:** Very Good
**Technical Quality:** Good
**Clarity Of Presentation:** Fair
**Impact:** 3

**Recommendation:**

Weak Accept: I recommend accepting the paper, but will not argue for my recommendation if the majority of other reviewers have a different opinion.

**Summary:**

This paper proposes a novel way to collaboratively perceive a scene through multiple robots that can work with asynchronous datasets. While prior work uses task specific losses to train bandwidth-limited encoders and decoders that handle message passing between robots, their method STAR learns to reconstruct the scene at each robot, and this scene can be used for downstream tasks. Additionally, training STAR is designed for asynchronously collected datasets, where robot messages are not aligned in time. STAR learns to encode and decode just the view of any single robot at a time, and then at inference time will use messages from each robot to reconstruct each robot’s current view. Then each view can be easily aggregated into one scene view. This scene view can be used for downstream task learning. They demonstrate their method achieves decent performance under much lower bandwidth with extensive experiments on an existing dataset.

**Issues:**

See “Weaknesses” for concerns for the authors to address.

Other types of task-agnostic feature learning

- Rather than learning to reconstruct the exact full scene, the authors should compare against methods that learn an invariant scene representation using multiple dowstream tasks, e.g. contrastive learning methods or meta-learning. This would help convince the reader that reconstructing the exact scene is a better choice for this intermediate representation.

Writing Style/Figure Changes

- It can be confusing to parse the Method Section, and was not immediately clear to me how the design desiderata in section 4.1 are incorporated in the method. I think the paper would greatly benefit from a method figure (possibly replacing the asynchronous vs. synchronous figure or some part of it) that shows the transformer design, the multi-robot-aggregation, and then downstream task prediction all in one figure. Then the authors could refer to this throughout section 4.
- The Temporal amortization section in 4.1 was quite confusing on first read, I think the issue in prior work could be more clearly stated or re-stated here and tied into STAR. Again having part of the method figure refer to the sub-sampling process would be quite helpful.
- Each desiderata in section 4.1 could benefit from more information, as a lot of details are glossed over here. For example in the Synchonization section, the authors say “Specifically, we use single-view observation as the supervision” without explaining how that supervision is employed during training.
- In the appendix, an algorithm outline would also be quite beneficial to outline the exact encoder / decoder steps during both training and inference.

**Quality Of The Limitations Section:**

Limitations are addressed clearly

**Reviewer Expertise:**

4: The reviewer is confident but not absolutely certain that the evaluation is correct

**Robotics Focus:**

Highly relevant to robotics but no hardware experiments

**Strengths And Weaknesses:**

Strengths

- Using temporal information to reduce bandwidth is a nice insight
- The scene reconstruction is a natural choice for a task-agnostic representation for ease of downstream task learning
- The authors run many ablations on their method, and compare against a good set of baselines.

Weaknesses

- At times, the writing style was confusing, with many critical details glossed over
- Conceptually, reconstructing the full scene from each robot’s messages seems like a much more challenging task than simply learning the downstream task. This is exemplified by the lower performance of STAR on the downstream tasks than task specific models, suggesting that the reconstructed scenes are not accurate enough for downstream task learning. While I agree task agnostic learning makes sense for quick transfer across tasks, the choice in using the full scene point cloud as that task agnostic representation is not convincing based on these results.
- Limited datasets: Only one task (one dataset) is considered, and it is a simulated task. It remains to be seen if STAR can perform well on datasets collected with real world hardware.

**Summary Of Recommendation:**

The contribution of the paper is quite novel and the experiments and ablations are mostly quite thorough. Although the chosen dataset is limited and simulated, other works in this field seem to compare against the same dataset.

---

> ### Author Response · Authors · 2022-08-26
> **Thank you sincerely for the helpful feedback!**
>
> **1. At times, the writing style was confusing, with many critical details glossed over.**
>
> Thanks for the comment. We have revised the writing to try to address this concern.
>
> **2. Conceptually, reconstructing the full scene from each robot’s messages seems like a much more challenging task than simply learning the downstream task. This is exemplified by the lower performance of STAR on the downstream tasks than task specific models, suggesting that the reconstructed scenes are not accurate enough for downstream task learning. While I agree task agnostic learning makes sense for quick transfer across tasks, the choice in using the full scene point cloud as that task agnostic representation is not convincing based on these results.**
>
> Thanks for the question. We have updated Table 1 and 3 with stronger results for both the scene reconstruction and the downstream tasks, and it now outperforms the CNN baselines. These improved results are due to a better choice of some hyperparameters, i.e., the transformer structure’s width and depth in our STAR methods. We believe with the updated results, your concerns are well addressed and our proposal for task agnostic representation becomes more convincing.
>
> Nevertheless, we agree that the full scene reconstruction task is more challenging and the task-specific models perform better on the respective downstream tasks. However, we would like to emphasize that the task-specific models are directly optimized with regards to the specific downstream task objectives. It means that following the task-specific approach *one needs to separately train a new collaborative model each time for a new task*. What’s more, it requires *carefully synchronized* data for each of the tasks. On the other hand, our proposed method allows for asynchronous training. We show the effectiveness of this method by showing the large improvement in performance when directly applying the trained *single robot perception model* on the reconstructed scenes **without additional training**. Therefore, besides the advantage of quick transferring to new tasks, the proposed method can have two **potential further improvements**:
>
> 1. Getting better **reconstruction quality** when exposed to **more larger-scale asynchronous training data**. This is especially meaningful in practice when synchronous training data is more costly to collect.
>
> 2. Getting better **performance** on the downstream tasks when applying a **stronger single robot perception model**.
>
> In the meantime, as the reviewer mentioned, we agree that exploring other choices as the task agnostic representation is an interesting idea and we plan to leave this as the future work.
>
>                 Table 3: The downstream tasks.
> | Method       | Det AP@IoU=0.5 | Det AP@IoU=0.7 | Seg Vehicle | Seg mIoU  |
> |--------------|----------------|----------------|-------------|-----------|
> | Lower-bound  | 49.90          | 44.21          | 45.93       | 36.64     |
> | STAR TS1     | **62.84**      | **57.22**      | **56.41**   | **39.09** |
> | STAR TS2     | **61.48**      | **55.75**      | **56.13**   | **38.97** |
> | VQ-VAE       | 60.27          | 54.08          | 55.40       | 38.48     |
> | CNN baseline | 59.85          | 54.05          | 54.61       | 38.32     |
> | Upper-bound  | **65.09**      | **60.26**      | **60.34**   | **40.45** |
>
>                 Table 1: The scene reconstruction. Updates are highlighted in bold font. Only results for spatial resolution 32 x 32 are listed here due to the space. Please see the Table 1 in our revised manuscript for the results of other resolutions.
> | Method       | IoU scale 1:1  | IoU scale 1:2 | IoU scale 1:4 | Communication Bandwidth |
> |--------------|----------------|---------------|---------------|-------------------------|
> | STAR TS1     | **55.13**      | **77.40**     | **83.28**     | 1.3 MB/s                |
> | STAR TS2     | **54.93**      | **75.71**     | **82.51**     | 640.0 KB/s              |
> | STAR TS3     | **53.35**      | **72.52**     | **79.20**     | 427.0 KB/s              |
> | STAR TS4     | **53.65**      | **72.98**     | **79.73**     | 320.0 KB/s              |
> | CNN backbone | 55.37          | 77.17         | 83.51         | 155.0 MB/s              |

---

> > ### Author Response · Authors · 2022-08-26
> > **Thank you sincerely for the helpful feedback!**
> >
> > **3. Limited datasets: Only one task (one dataset) is considered, and it is a simulated task. It remains to be seen if STAR can perform well on datasets collected with real world hardware.**
> >
> > We employ the large-scale V2X-Sim [1] in our experiments since it supports different perception tasks, which could help us evaluate our task-agnostic collaborative perception framework.
> >
> > [1] Li, Yiming, Dekun Ma, Ziyan An, Zixun Wang, Yiqi Zhong, Siheng Chen, and Chen Feng. "V2X-Sim: Multi-Agent Collaborative Perception Dataset and Benchmark for Autonomous Driving." IEEE Robotics and Automation Letters (2022).
> >
> > ---
> >
> > **4. Other types of task-agnostic feature learning. Rather than learning to reconstruct the exact full scene, the authors should compare against methods that learn an invariant scene representation using multiple downstream tasks, e.g. contrastive learning methods or meta-learning. This would help convince the reader that reconstructing the exact scene is a better choice for this intermediate representation.**
> >
> > Thank you for this interesting suggestion. In this work, we employ autoencoding to learn the shared representations because its output could  be used directly in the downstream tasks: the reconstructions could be seamlessly utilized by off-the-shelf individual perception model trained on single-view data without any fine-tuning, bridging the gap between collaborative perception and individual perception. However, for meta-learning or contrastive learning, it is non-trivial to apply them into our task-agnostic framework. We agree with you that this is a very interesting topic for future research and we welcome any insights or suggestions.
> >
> > ---
> >
> > **5. Writing Style/Figure Changes**
> >
> > **It can be confusing to parse the Method Section, and was not immediately clear to me how the design desiderata in Section 4.1 are incorporated in the method. I think the paper would greatly benefit from a method figure (possibly replacing the asynchronous vs. synchronous figure or some part of it) that shows the transformer design, the multi-robot-aggregation, and then downstream task prediction all in one figure. Then the authors could refer to this throughout section 4.**
> >
> > Thank you for the suggestions.
> > In the method section we aim to describe a general framework (MRSC) that is independent from downstream tasks and is trainable from asynchronous datasets. Then Section 4 describes a specific solution. We present the design desideratas with two ideas in mind: (1) it fits the framework we proposed in Section 3, allowing asynchronous training and synchronous inference; (2) it achieves excellent performance-bandwidth tradeoff as this is an important consideration in previous collaborative perception works.
> >
> > We omit the specific transformer design in the method figure because the structure of transformer blocks are well explained in many previous works such as [1]. So we decided to follow the illustration figure in MAE and abstract the encoder and decoder as simple blocks for conciseness. In section 5.1 we also specified the exact width and depth for both the encoder and the decoder for your reference. Thank you for the suggestion. We have updated Figure 2 with illustrations of the multi-robot aggregation and the downstream tasks. Following the notations and equations introduced in Section 3. Please see our updated paper  for details.
> >
> > **The Temporal amortization section in 4.1 was quite confusing on first read, I think the issue in prior work could be more clearly stated or re-stated here and tied into STAR. Again having part of the method figure refer to the sub-sampling process would be quite helpful.**
> >
> > Thank you for the suggestions. We have revised the temporal amortization part in section 4.1 to provide a more detailed explanation.
> > The issues in the prior works are discussed mainly in the introduction, we have modified Section 3 and restated the issues that motivate this work. Please see the “Motivation” part in section 3 for details.
> >
> > **Each desiderata in section 4.1 could benefit from more information, as a lot of details are glossed over here. For example in the Synchronization section, the authors say “Specifically, we use single-view observation as the supervision” without explaining how that supervision is employed during training.**
> >
> > Thank you for the suggestion, we have revised Section 4.1 providing more detail to avoid information being glossed over. Also, in Section 4.3 more information about the training and inference is provided.
> >
> > **In the appendix, an algorithm outline would also be quite beneficial to outline the exact encoder / decoder steps during both training and inference.**
> >
> > Thank you for the suggestion. We have included an algorithm outline in the supplementary hoping that could improve the clarity of the presentation. We will release the codes when this work gets accepted.
> >
> > [1] Dosovitskiy, Alexey, et al. "An image is worth 16x16 words: Transformers for image recognition at scale.". ICLR, 2020.

---

### Official Review · Reviewer_8X6f · 2022-07-31

**Originality:** Good
**Technical Quality:** Good
**Clarity Of Presentation:** Very Good
**Impact:** 2

**Recommendation:**

Weak Accept: I recommend accepting the paper, but will not argue for my recommendation if the majority of other reviewers have a different opinion.

**Summary:**

This paper proposes a self-supervised learning strategy based on multi-robot scene completion, for which multiple robots collaboratively reconstruct the full view based on each robot’s partial observation. The training is built on top of a MAE type training strategy, for which the data can be asynchronized collected from different robots.

**Issues:**

See weaknesses.

**Quality Of The Limitations Section:**

Limitations are addressed clearly

**Reviewer Expertise:**

2: The reviewer is willing to defend the evaluation, but it is quite likely that the reviewer did not understand central parts of the paper

**Robotics Focus:**

Relevant but unlikely to deploy to hardware in near future

**Strengths And Weaknesses:**

Disclaimer: I am not the expert in this area.

Strength. The paper proposes to apply a spatial-temporal MAE for efficient task-agnostic multi-robot scene completion. The proposed method has achieved good performance on downstream detection and segmentation tasks.

Limitations.

-- Contribution. Though I understand the general design strategy of applying spatial-temporal MAE for asynchronous training, I am not following how the proposed method is more advanced/better compared to other baseline methods. The authors did mention that the proposed method requires a much lower bandwidth compared to CNN counter-part, but it limits the reconstruction quality as shown in Table 1.

-- Memory. The proposed method is built on top of MAE, which should require a much large memory compared to CNN baseline. Would the memory be the bottleneck for communication as well?


**Summary Of Recommendation:**

The paper presents an interesting framework for collaborative scene completion based on MAE. I feel like the design fits human intuition, though the experiments can be improved in terms of clarity -- 1.) adding a small introduction to each task-specific baselines.  ii) Explain why task-specific methods can achieve even higher performance than task-agnostic upper-bounds. iii) Additional insights and advantages compared to other designs.

---

> ### Author Response · Authors · 2022-08-26
> **Thank you sincerely for the constructive feedback!**
>
> **1. Contribution. Though I understand the general design strategy of applying spatial-temporal MAE for asynchronous training, I am not following how the proposed method is more advanced/better compared to other baseline methods. The authors did mention that the proposed method requires a much lower bandwidth compared to CNN counterpart, but it limits the reconstruction quality as shown in Table 1.**
>
> Thanks for the question. We would like to update the experiment results of STAR in both Table 1 and Table 3 in the manuscript. After the initial submission, we have explored more choices of the transformer structure, including the hidden dimensions and the depths of the encoder and decoder. Our current results outperform the CNN baseline in both scene reconstruction quality and the performances on the downstream tasks. We have attached the updated results in the tables below for your convenience. Please see Table1 and 3 in our revised manuscript for the complete results.
>
>                      Table 3: The downstream tasks.
> | Method       | Det AP@IoU=0.5 | Det AP@IoU=0.7 | Seg Vehicle | Seg mIoU  |
> |--------------|----------------|----------------|-------------|-----------|
> | Lower-bound  | 49.90          | 44.21          | 45.93       | 36.64     |
> | STAR TS1     | **62.84**      | **57.22**      | **56.41**   | **39.09** |
> | STAR TS2     | **61.48**      | **55.75**      | **56.13**   | **38.97** |
> | VQ-VAE       | 60.27          | 54.08          | 55.40       | 38.48     |
> | CNN baseline | 59.85          | 54.05          | 54.61       | 38.32     |
> | Upper-bound  | **65.09**      | **60.26**      | **60.34**   | **40.45** |
>
>                     Table 1: The scene reconstruction. Updates are highlighted in bold font. Only results for spatial resolution 32 x 32 are listed here due to the space. Please see the Table 1 in our revised manuscript for the results of other resolutions.
> | Method       | IoU scale 1:1  | IoU scale 1:2 | IoU scale 1:4 | Communication Bandwidth |
> |--------------|----------------|---------------|---------------|-------------------------|
> | STAR TS1     | **55.13**      | **77.40**     | **83.28**     | 1.3 MB/s                |
> | STAR TS2     | **54.93**      | **75.71**     | **82.51**     | 640.0 KB/s              |
> | STAR TS3     | **53.35**      | **72.52**     | **79.20**     | 427.0 KB/s              |
> | STAR TS4     | **53.65**      | **72.98**     | **79.73**     | 320.0 KB/s              |
> | CNN backbone | 55.37          | 77.17         | 83.51         | 155.0 MB/s              |
>
> ---
>
> **2. Memory. The proposed method is built on top of MAE, which should require a much larger memory compared to CNN baseline. Would memory be the bottleneck for communication as well?**
>
> Thank you for the question. We admit that the memory constraint will indeed bring challenges when designing efficient robotics systems. However, we only aim to discuss the trade off between perception performance and communication burden in this work, and this is not directly related to the memory constraint because we are NOT transmitting all the intermediate features (which has a large runtime memory footprint locally in each robot’s computer) through a communication network, instead we only need to transmit the final layer’s feature from our encoder which has a small memory footprint, therefore memory is not the bottleneck for communication. As for more practical concerns such as large local memory footprints might slow down the communication, we find it to be out of the scope of this research and requires more engineering discussions specific to particular hardware realization of the proposed collaborative perception system.
>
> Besides, we would also like to note that we have updated our results with a lighter version of the STAR model compared to the initial submission, it also obtained better results. Please see Table 1 and 3 for more details.

---

> > ### Author Response · Authors · 2022-08-26
> > **Thank you sincerely for the constructive feedback!**
> >
> > **3. I feel like the design fits human intuition, though the experiments can be improved in terms of clarity -- 1.) adding a small introduction to each task-specific baselines. ii) Explain why task-specific methods can achieve even higher performance than task-agnostic upper-bounds. iii) Additional insights and advantages compared to other designs.**
> >
> > Thank you for the question.
> > For i), we decided to omit the introductions to each task-specific baseline in the initial submission from the main paper since the space is limited and readers can find those in published works. We can also add those in the appendix/supplementary materials of the final paper to make our paper more self-contained. The reviewer can refer to these works for a better understanding. We have also revised Section 5, adding a paragraph of introductions hoping that this is helpful for the presentation of this work.
> >
> > For ii), in particular, the single-robot perception model in the upper-bound is trained only using single-robot scene observation. No more supervision on the multi-view observation is involved. While the task-specific models benefit from the direct supervision on the multi-view observation of the specific downstream tasks. This can result in the difference in performance.
> >
> > For iii) compared to the previous task-specific approaches, this design enables asynchronous training while synchronous inference. **It can bridge the gap between single-robot perception and multi-robot perception**, and achieve excellent performance-bandwidth tradeoff. It provides the insight that with collaborative task-agnostic representation and single-robot perception model, *one can build effective collaborative perception systems that are compatible with more downstream tasks without being constrained by carefully collecting and annotating well synthesized multi-robot data*, which is more costly than those for single-robot perception.

---

### Official Review · Reviewer_MFnL · 2022-08-03

**Originality:** Good
**Technical Quality:** Fair
**Clarity Of Presentation:** Fair
**Impact:** 2

**Recommendation:**

Weak Reject: I recommend rejecting the paper, but will not argue for my recommendation if the majority of other reviewers have a different opinion.

**Summary:**

This paper tackles the task of ‘collaborative’ scene completion, where multiple robots communicate their own observations (top-down maps) so that each can build a more complete map. Subsequent tasks e.g. detection/segmentation can then be performed on such a completed map.

The key technical question this work addresses is regarding the representation of a robot’s own map that it should communicate. Inspired by MAEs, this work shows that a small number of patch embeddings allow decoding a map (and trains a MAE-like encoder and decoder to predict occupancy maps). One innovation here is that the paper extends MAEs to possibly use patch embeddings from previous timesteps (this helps reduce the communication at each timestep, as only transmitting a subset of embeddings suffices).

The paper reports scene completion results on outdoor driving datasets and shows that the proposed approach leads to accurate reconstructions despite low-bandwidth communications.

**Issues:**

- Please do correct me if I am misinterpreting the ‘independent’ nature of the per-robot map reconstructions.

- I would be curious to hear the authors’ thoughts on why alternate compression schemes e.g. VQ-VAEs would be less preferred compared to the method here of using a MAE-like network.


**Quality Of The Limitations Section:**

Limitations are addressed clearly

**Reviewer Expertise:**

4: The reviewer is confident but not absolutely certain that the evaluation is correct

**Robotics Focus:**

Relevant but unlikely to deploy to hardware in near future

**Strengths And Weaknesses:**

Strengths:

a) The overall task studied is a well-motivated one i.e. how should agents communicate to effectively reconstruct a scene while minimizing the information transmitted, and I have not seen prior works which explore this question in detail.

b) The proposed approach to learn a MAE-like network that. can leverage embeddings across timestamps is a nice contribution. In particular, it helps with temporal amortization of communication and is shown to only result in slight performance drops (compared to the communication efficiency gains).

Weaknesses:

a) For a paper claiming to be about ‘collaborative’ completion, I feel the approach is anything but collaborative. Given the asynchronous training, it seems the final training procedure is to simply learn a MAE-like network (albeit with possible temporal data indicating past views from the same robot) to independently reconstruct the scene for each robot given only the observations from the perspective of that robot. While the MAE-like training can help get a lower bandwidth representation of the map (i.e. in terms of a subset of transformer features), this completely sidesteps the ‘collaborative’ nature of the task (e.g. robot one may have observed one end of a long table, robot 2 may have observed the other end, and a true collaborative approach would leverage both to possibly hallucinate the middle!).

b) Several important details are missing, and in its current form, it is rather difficult to parse what the work exactly does! In particular:
- what precisely is $\Gamma$ in L177?
- The details of the STAR decoder are not explained well. L169-L177 seem to imply that this uses tokens (output by the encoder, possibly across timesteps) to then predict a mask — but it is not clear what fraction of tokens are masked.
- what does the CNN backbone baseline in Table 1 actually correspond to?

c) If the main goal is to reduce the dimensionality of the messages (so that the one robot can decode the maps sent by others), why not leverage something akin to a VQ-VAE which, using discrete tokens, significantly reduces the communication bandwidth? For example, given a vocabulary of 512 and a spatial resolution of 32 x 32 for the tokens, the size of each message would be around 1 KB — leading to a significantly smaller bandwidth! Again, if the paper were actually doing some collaborative completion, this would bot have been as issue, but in its current form, it is only searching for ways to effectively compress the maps observed by each robot so they can be communicated and (independently) decoded.

d) I am also not convinced regarding the presentation of the work as ‘task-agnostic perception’. In fact, there is a very specific defined task that the network is trained for (i.e. MAE-like completion). Given a completed scene, one does need to use separately trained networks to perform any desired task (like detection, segmentation). This would be akin to claiming that ‘image completion’ is ’task agnostic visual understanding’ as given the completed image,  one can use off-the-shelf systems to do later tasks.

e) On a note related to the ‘multi-robot’ application setting, the work here actually abstracts away the actions the robots must take (which is a rather key part of a multi-robot. setting i.e. how should agents act). This work actually treats ‘robots’ as  as LiDAR sensors that passively move, and I am not sure this is a very good setup to investigate.


**Summary Of Recommendation:**

First and foremost, I believe this work does not tackle ‘collaborative completion’ as each robot’s observations are independently processed into a corresponding map (which are then trivially combined) (W1). The primary benefit of the proposed approach is a more compressed representation of a map s.t. it can be communicated easily, but there alternate methods like VQ-VAEs would perhaps be even better (W3). Further, there are also concerns regarding the presentation (W2) and the positioning (W4).

While the temporal amortization of communication is an interesting and novel nugeet, I think the above concerns far outweigh this aspect.

---

> ### Author Response · Authors · 2022-08-26
> **Thank you sincerely for the insightful comments!**
>
> **(a) For a paper claiming to be about ‘collaborative’ completion, I feel the approach is anything but collaborative. Given the asynchronous training, it seems the final training procedure is to simply learn a MAE-like network (albeit with possible temporal data indicating past views from the same robot) to independently reconstruct the scene for each robot given only the observations from the perspective of that robot. While the MAE-like training can help get a lower bandwidth representation of the map (i.e. in terms of a subset of transformer features), this completely sidesteps the ‘collaborative’ nature of the task (e.g. robot one may have observed one end of a long table, robot 2 may have observed the other end, and a true collaborative approach would leverage both to possibly hallucinate the middle!)**
>
>
> **A:** Thank you for the great question. Regarding the concern of describing our method as “collaborative”, we are following the established usage in collaborative perception literature [1-5], in which the term “collaborative” indicates **information exchange** amongst multiple robots to collectively perceive a scene. In our approach, the robot can utilize the information *shared by others* to reconstruct a complete scene viewed by all robots. Therefore, we describe our task & approach using the term “collaborative”.
>
> Additionally, please note that the proposed “collaborative completion” could be trained in both **synchronous** and **asynchronous** modes (see line 139-144). For the synchronous mode, both training and inference involves collaboration as defined above. For the asynchronous mode, during inference, multiple robots use the same encoder to independently convert each individual view into a feature representation, and then share the feature with teammates for the same decoder to complete the scene in parallel. Therefore, although during asynchronous training no information is exchanged between agents, at the inference time, there is a clear exchange of information (the shared features) to complete the scene. *Thus, whether using synchronous or asynchronous training, we find it proper to call our method collaborative completion.*
>
> Regarding your example requiring hallucination, we totally agree with you that this is useful to demonstrate collaboration benefits. **Indeed, such hallucination could be learned naturally in our synchronous training.** We have added this point in Section 3 (see line 144-147), where we propose a general framework of collaborative scene completion. As for the specific realization of the framework in Section 4, we employ the **simple yet effective** asynchronous mode to leverage abundant single-robot recordings for training collaborative perception deep networks, which is novel and useful in practice for the multi-robot collaborative perception community which suffers from the *lack of large-scale real-world synchronized multi-robot recordings (and annotations)*. And as mentioned above, adding this new training mode does not remove collaboration during inference.
>
> In summary:
> 1. “Collaborative” actually denotes **information exchange** amongst multiple robots, following the terminology in collaborative perception literature.
> 2. The proposed collaborative scene completion is a **general formulation** supporting both synchronous and asynchronous training. We proposed asynchronous mode since it **solves some practical challenges** in the current community.
> 3. We do not require our model to hallucinate regions of a scene that are not observed by any robots in the team, although such a **collaborative completion with hallucination** could be achieved by synchronous training with sufficiently complete scene structures serving as the reconstruction targets.
>
> [1] Li, Yiming, Dekun Ma, Ziyan An, Zixun Wang, Yiqi Zhong, Siheng Chen, and Chen Feng. "V2X-Sim: Multi-Agent Collaborative Perception Dataset and Benchmark for Autonomous Driving." IEEE Robotics and Automation Letters (2022).
>
> [2] Xu, Runsheng, Hao Xiang, Xin Xia, Xu Han, Jinlong Li, and Jiaqi Ma. "Opv2v: An open benchmark dataset and fusion pipeline for perception with vehicle-to-vehicle communication." In ICRA, pp. 2583-2589. IEEE, 2022.
>
> [3] Liu, Yen-Cheng, Junjiao Tian, Chih-Yao Ma, Nathan Glaser, Chia-Wen Kuo, and Zsolt Kira. "Who2com: Collaborative perception via learnable handshake communication." In 2020 IEEE International Conference on Robotics and Automation (ICRA), pp. 6876-6883. IEEE, 2020.
>
> [4] Wang, Tsun-Hsuan, Sivabalan Manivasagam, Ming Liang, Bin Yang, Wenyuan Zeng, and Raquel Urtasun. "V2vnet: Vehicle-to-vehicle communication for joint perception and prediction." In European Conference on Computer Vision, pp. 605-621. Springer, Cham, 2020.
>
> [5] Li, Yiming, Shunli Ren, Pengxiang Wu, Siheng Chen, Chen Feng, and Wenjun Zhang. "Learning distilled collaboration graph for multi-agent perception." Advances in Neural Information Processing Systems 34 (2021): 29541-29552.

---

> > ### Author Response · Authors · 2022-08-26
> > **Thank you sincerely for the insightful comments!**
> >
> > **(b) Several important details are missing, and in its current form, it is rather difficult to parse what the work exactly does! In particular:
> > What precisely is Γ in L177?**
> >
> >  Thanks for pointing this out. We add more details in our paper, see line 126-131.
> >
> > **The details of the STAR decoder are not explained well. L169-L177 seems to imply that this uses tokens (output by the encoder, possibly across timesteps) to then predict a mask — but it is not clear what fraction of tokens are masked.**
> >
> > The patch tokens are randomly masked in a given timestamp. And the masked patches are spatially complementary across the temporal communication period. For example, considering an observed BEV region being patchified into 2x2 patch [A,B; C,D] in two consecutive timestamps t=0 and 1, if we randomly mask out B and C at t=0, then we can mask out A and D at t=1; and if we randomly mask out A and B at t=0, then we can mask out C and D at t=1; etc. In this way, we ensure that the remaining patches at t=0 and t=1 will jointly cover the entire spatial region (although at different timestamps) and the model learns to reconstruct the whole scene structure at each timestamp.
> >
> > **What does the CNN backbone baseline in Table 1 actually correspond to?**
> >
> > As we mentioned in section 5.1, we used the FaFNet [1] as the CNN backbone baseline since it provides strong results in downstream tasks and is a popular baseline choice in the task-specific works. We add more details about our baseline methods in the experimental setup (see line 239-249).
> >
> > [1] Luo, Wenjie, Bin Yang, and Raquel Urtasun. "Fast and furious: Real time end-to-end 3d detection, tracking and motion forecasting with a single convolutional net." Proceedings of the IEEE conference on Computer Vision and Pattern Recognition. 2018.
> >
> > ---
> >
> > **(c) If the main goal is to reduce the dimensionality of the messages (so that the one robot can decode the maps sent by others), why not leverage something akin to a VQ-VAE which, using discrete tokens, significantly reduces the communication bandwidth? For example, given a vocabulary of 512 and a spatial resolution of 32 x 32 for the tokens, the size of each message would be around 1 KB — leading to a significantly smaller bandwidth! Again, if the paper were actually doing some collaborative completion, this would not have been an issue, but in its current form, it is only searching for ways to effectively compress the maps observed by each robot so they can be communicated and (independently) decoded.**
> >
> > Thank you for this great question. First of all, as explained above, collaborative hallucination could indeed be achieved in our synchronous collaboration mode, therefore it is not “an issue” anymore as you pointed out and there is no need to compare with VQ-VAE-like baselines.
> > **However, we find your suggestion on VQ-VAE very insightful, especially if we do not require the method to hallucinate the regions not observed at all, such as in our asynchronous mode.** Therefore, we took your suggestion and implemented a VQ-VAE baseline, as shown below.
> > Please see our revised manuscript for the complete quantitative results:
> > | Method       | Det AP@IoU=0.5 | Det AP@IoU=0.7 | Seg Vehicle | Seg mIoU  |
> > |--------------|----------------|----------------|-------------|-----------|
> > | Lower-bound  | 49.90          | 44.21          | 45.93       | 36.64     |
> > | STAR TS1     | **62.84**      | **57.22**      | **56.41**   | **39.09** |
> > | STAR TS2     | **61.48**      | **55.75**      | **56.13**   | **38.97** |
> > | VQ-VAE       | 60.27          | 54.08          | 55.40       | 38.48     |
> > | CNN baseline | 59.85          | 54.05          | 54.61       | 38.32     |
> > | Upper-bound  | **65.09**      | **60.26**      | **60.34**   | **40.45** |
> >
> > From this new baseline, we can see that **VQ-VAE baseline is indeed another effective realization of our framework of asynchronous collaborative completion (without hallucination)**, with slightly worse (or comparable) performance than our STAR realization. We wholeheartedly thank you for suggesting such a simple and effective realization and we will make sure to properly acknowledge your idea in our final paper.

---

> > > ### Author Response · Authors · 2022-08-26
> > > **Thank you sincerely for the insightful comments!**
> > >
> > > **(d) I am also not convinced regarding the presentation of the work as ‘task-agnostic perception’. In fact, there is a very specific defined task that the network is trained for (i.e. MAE-like completion). Given a completed scene, one does need to use separately trained networks to perform any desired task (like detection, segmentation). This would be akin to claiming that ‘image completion’ is ’task agnostic visual understanding’ as given the completed image, one can use off-the-shelf systems to do later tasks.**
> > >
> > > Thank you for raising this question about our choice of words. We are following the **common terminology** in the self-supervised representation learning community [1,4,5] so as to achieve a good balance between technical clarity and conciseness in our writing. We add more explanations to avoid confusions (see line 71-75, 86-101 in our updated paper), and we are open to suggestions on better choices of expressions. Nevertheless, the following is our detailed thoughts and reasons for the current choice of words:
> > >
> > > 1. The dictionary definition of the word **“perception”** is “the state of being or process of becoming aware of something through the senses”. In our work, multiple agents each sense a small local area of the environment, then share with each other their sensed information, eventually collaboratively becoming aware of the 3D shape of the larger common areas. Although this does not directly generate common perception task outputs like object boxes, trackets, or segmentation masks, which is also why we use the word “task-agnostic”, it still satisfies the above dictionary definition of perception.
> > >
> > > 2. As mentioned earlier, according to the literature, **“collaborative perception”** means perception through information exchange/sharing between multiple agents. In our task, the agents indeed perceive the 3D surroundings more comprehensively via collaboration. Therefore, using “collaborative perception” to describe our method is proper and easy to understand by our readers.
> > >
> > > 3. Perception does not only mean to detect or segment objects. Perceiving the 3D environment more comprehensively than a single agent is also a non-trivial perception task. Different from the image reconstruction task in the regular image denoising autoencoder which is low-level image processing, our **collaborative reconstruction/completion** task can be directly used in higher-level robotics applications such as path planning or collision avoidance, while image completion or denoising cannot. Therefore, we find the analogy between our task to image completion inaccurate.
> > >
> > > 4. **Task-agnostic collaborative perception**. In this work, we define task-agnostic collaborative perception as the intermediate collaborative perception sharing **task-independent** representations amongst multiple robots, which is different from the existing task-specific collaborative perception sharing **task-dependent** representations. The aforementioned “task” is defined as the **downstream task** the system actually aims to solve: robot perception tasks such as object detection, object tracking, and semantic segmentation. In contrast, scene reconstruction is the **pretext task** (well-known in the self-supervised learning community [1]) which is designed to learn a compact representation to share amongst robots without the requirement of laborious human annotations.
> > >
> > > 5. **An analogy between our work and classic self-supervised learning.** For better understanding, let us review two primary components in self-supervised learning workflow [1]: (1) **self-supervised pre-training**: pre-training a model using a "generic" dataset (such as ImageNet) that does not correspond to the task you need to solve but enables the model to learn some "general" representations with self-supervision from carefully-designed pretext tasks such as contrastive learning [2] or patch reconstruction [3], and (2) **downstream task adaptation**: use the dataset that defines the target problem we are trying to solve to fine-tune this pre-trained model. **In our case, the shared representations are learned via the defined pretext task scene completion,  and the representations are not fine-tuned on the downstream perception**: the completed results reconstructed with the shared features are directly fed into the well-developed single-robot perception models. Such kind of (downstream) task-independent representation learning is generally described by “task-agnostic” in the machine learning community [4,5].

---

> > > > ### Author Response · Authors · 2022-08-26
> > > > **Thank you sincerely for the insightful comments!**
> > > >
> > > > In summary, the “task” in task-agnostic collaborative perception denotes the **downstream perception task**, and the scene reconstruction task is actually the **pretext task** for self-supervised representation learning.
> > > > We claim our framework as task-agnostic since the shared representations are **task-independent**, which is different from existing collaborative perception frameworks which employ **task-dependent** feature representations as the communication messages.
> > > >
> > > > [1] Ericsson, Linus, Henry Gouk, Chen Change Loy, and Timothy M. Hospedales. "Self-Supervised Representation Learning: Introduction, advances, and challenges." IEEE Signal Processing Magazine 39, no. 3 (2022): 42-62.
> > > >
> > > > [2] He, Kaiming, Haoqi Fan, Yuxin Wu, Saining Xie, and Ross Girshick. "Momentum contrast for unsupervised visual representation learning." In Proceedings of the IEEE/CVF conference on computer vision and pattern recognition, pp. 9729-9738. 2020.
> > > >
> > > > [3] He, Kaiming, Xinlei Chen, Saining Xie, Yanghao Li, Piotr Dollár, and Ross Girshick. "Masked autoencoders are scalable vision learners." In Proceedings of the IEEE/CVF Conference on Computer Vision and Pattern Recognition, pp. 16000-16009. 2022.
> > > >
> > > > [4] Evain, Solène, Manh Ha Nguyen, Hang Le, Marcely Zanon Boito, Salima Mdhaffar, Sina Alisamir, Ziyi Tong et al. "Task agnostic and task specific self-supervised learning from speech with LeBenchmark." In the Thirty-fifth Conference on Neural Information Processing Systems. 2021.
> > > >
> > > > [5] Nguyen, A. Tuan, Ser Nam Lim, and Philip Torr. "Task-Agnostic Robust Representation Learning." arXiv preprint arXiv:2203.07596 (2022).
> > > >
> > > > ---
> > > >
> > > > **(e) On a note related to the ‘multi-robot’ application setting, the work here actually abstracts away the actions the robots must take (which is a rather key part of a multi-robot. setting i.e. how should agents act). This work actually treats ‘robots’ as LiDAR sensors that passively move, and I am not sure this is a very good setup to investigate.**
> > > >
> > > > We agree with you that **active multi-robot perception** is a very interesting and important future topic. However, we choose the current setup because multi-robot perception research has just begun and there are still rooms for improvement, as shown in recent works in this research direction [1-7]. We believe even though we do not have actions in the current loop, this research is still a good fit for CoRL just like other common single-agent and non-active perception research such as 3D object detection [8].
> > > >
> > > > [1] Li, Yiming, Dekun Ma, Ziyan An, Zixun Wang, Yiqi Zhong, Siheng Chen, and Chen Feng. "V2X-Sim: Multi-Agent Collaborative Perception Dataset and Benchmark for Autonomous Driving." IEEE Robotics and Automation Letters (2022).
> > > > [2] Xu, Runsheng, Hao Xiang, Xin Xia, Xu Han, Jinlong Li, and Jiaqi Ma. "Opv2v: An open benchmark dataset and fusion pipeline for perception with vehicle-to-vehicle communication." In 2022 International Conference on Robotics and Automation (ICRA), pp. 2583-2589. IEEE, 2022.
> > > >
> > > > [3] Liu, Yen-Cheng, Junjiao Tian, Chih-Yao Ma, Nathan Glaser, Chia-Wen Kuo, and Zsolt Kira. "Who2com: Collaborative perception via learnable handshake communication." In 2020 IEEE International Conference on Robotics and Automation (ICRA), pp. 6876-6883. IEEE, 2020.
> > > >
> > > > [4] Liu, Yen-Cheng, Junjiao Tian, Nathaniel Glaser, and Zsolt Kira. "When2com: Multi-agent perception via communication graph grouping." In Proceedings of the IEEE/CVF Conference on computer vision and pattern recognition, pp. 4106-4115. 2020.
> > > >
> > > > [5] Wang, Tsun-Hsuan, Sivabalan Manivasagam, Ming Liang, Bin Yang, Wenyuan Zeng, and Raquel Urtasun. "V2vnet: Vehicle-to-vehicle communication for joint perception and prediction." In European Conference on Computer Vision, pp. 605-621. Springer, Cham, 2020.
> > > >
> > > > [6] Li, Yiming, Shunli Ren, Pengxiang Wu, Siheng Chen, Chen Feng, and Wenjun Zhang. "Learning distilled collaboration graph for multi-agent perception." Advances in Neural Information Processing Systems 34 (2021): 29541-29552.
> > > >
> > > > [7] Vadivelu, Nicholas, Mengye Ren, James Tu, Jingkang Wang, and Raquel Urtasun. "Learning to Communicate and Correct Pose Errors." In Conference on Robot Learning, pp. 1195-1210. PMLR, 2021.
> > > >
> > > > [8] Wang, Yue, Vitor Campagnolo Guizilini, Tianyuan Zhang, Yilun Wang, Hang Zhao, and Justin Solomon. "Detr3d: 3d object detection from multi-view images via 3d-to-2d queries." In Conference on Robot Learning, pp. 180-191. PMLR, 2022.

---

> > > > > ### Comment · Reviewer_MFnL · 2022-08-27
> > > > > **Thanks for the response**
> > > > >
> > > > > I’d like to that the authors for the response, and in particular for adding several missing details and reporting the added VQ-VAE experiments. Unfortunately, my concerns still persist.
> > > > >
> > > > > Collaboration:
> > > > > While the proposed method ‘could’ be trained synchronously, it is not in the current version. Similarly, the collaboration during inference, as described in the updated manuscript in L210 is simply averaging of the (independent) per-robot predictions.
> > > > >
> > > > >
> > > > > Alternate Compression:
> > > > > In the asynchronous training, the approach essentially learns to complete maps given a subset of tokens (as in MAEs), and the primary benefit of this (compared to the “upper bound” where robots can communicate all observations) is the compression of the communication. As the VQ-VAE ablation (where robots simply send each other a compressed map via VQ-VAE discretization) shows, the existing alternate is already competitive in performance (the bandwidth for this is not reported unfortunately, but I assume this is more efficient than the proposed method due to the discretization).

---

> > > > > > ### Author Response · Authors · 2022-08-27
> > > > > > **Thank you very much for the timely feedback!**
> > > > > >
> > > > > > We highly appreciate your timely feedback, and we really enjoy the discussions with you. We have some different opinions for your new comments, and we kindly ask you to rethink the overall picture and our motivation. Here are our detailed responses:
> > > > > >
> > > > > > 1. As we mentioned a lot of times, asynchronous training is **more valuable** than synchronous training, because the multi-robot collaborative perception community suffers from *the lack of large-scale real-world synchronized multi-robot recordings (and annotations)*. And we for the first time propose to **use asynchronous learning to solve collaborative perception problems**. Synchronous training is **NOT THE FOCUS** of this paper, as we clearly mentioned in our motivation (line 26-31; line 179-187).
> > > > > > Meanwhile, we respectfully disagree with you that simplicity is our weakness: *we strongly believe that simple yet effective methods should be more appreciated than ones with bells and whistles, in both academia and industry*: **only with such a simple framework, the performance of existing single-robot detector can be improved by 23.2% (49.9 -> 61.48 on AP@0.5) without ANY additional training**!
> > > > > >
> > > > > > ---
> > > > > >
> > > > > > 2. We would like to remind you that we are investigating the **performance-bandwidth trade-off** but not only bandwidth itself. Despite higher bandwidth, STAR has better performance than VQ-VAE, e.g., the detection performance at IoU=0.7 is **improved by 3.1%** (55.75 vs. 54.08). Meanwhile, our high-level idea of temporal amortization is **NOT incompatible** with discrete quantization in VQ-VAE, and they are just two **different realizations** of our proposed framework. Actually, these two techniques can be jointly implemented for better results, and we are still the first to apply VQ-VAE in multi-robot perception (thank you again for the brilliant idea!).
> > > > > >
> > > > > > ---
> > > > > >
> > > > > > 3. Finally, we would like to remind you that we are the first to investigate this novel task of self-supervised collaborative perception, and we believe that our work can **inspire other researchers**, especially that *we are the first to try to bridge the gap between single-robot and multi-robot perception*. We strongly believe that the *high-level principle*, **TRAINING on single robot and TESTING on multiple robots**, is much more important than the *specific number* of performances, especially in the situation that both two methods have their own merits.
> > > > > >
> > > > > > Thank you again for your quick responses! We are open to any new question.

---

### Meta-Review · Area_Chair_83W4 · 2022-08-12

**Recommendation:** Accept (Poster)
**Confidence:** 4

**Metareview:**

The paper presents a method for multiple robots to communicate among themselves so that each one can build a spatial map using information from the others.  The focus is on efficient information sharing through the spatial-temporal-aware auto-encoder (STAR) concept.

All the reviewers agree that this is an important problem in robotics, and appreciate the focus on bandwidth constrained communication and asynchronous training.  The authors have addressed most of the reviewers' original concerns through supplemental material and revisions to the paper.

I encourage the authors to simplify their language and avoid jargon to better communicate their results to the CoRL audience.  Simply put, the robots in this work build an occupancy map using a LIDAR sensor and information communicated from neighbors.  Describing the contribution with clear, concrete language in the abstract and intro will help improve the ultimate impact of the paper.

---

> ### Author Response · Authors · 2022-08-26
> **Response to meta review.**
>
> We sincerely thank all the reviewers for their thorough and insightful comments and suggestions. All reviewers find novelties in our proposed framework, and most of the review questions were about **choice of words** or **writing clarification** which are addressed now.  Considering the **novelty** and **practical utility** of the proposed framework and method, we believe this paper is worthwhile to be accepted at CoRL since we are the first method to enable **TRAINING on single robot and TESTING on multiple robots**, which is a new paradigm in collaborative perception. We summarize our response based on the meta review in the following:
>
> 1. The reviewers are concerned that the results are not strong enough. We have uploaded **better results** after exploring more choices of hyperparameters of the transformer structure, including the hidden dimensions and the depths of the encoder and decoder. We have also revised the implementation details in the paper to list these changes. In the meantime, we have also explained the essential differences between our method and the task-specific ones that aim directly on the certain perception tasks and why this can lead to a difference in performance.
>
> 2. We have provided **more descriptions** of our proposed methods and used techniques both in the responses and in the revised manuscript, hoping that this will resolve the issue that many details were glossed over as reflected by the reviewers.
>
> 3. Regarding the concerns of reviewer MFnL, we elaborate our motivations, the problem setup and the important concepts such as “task-agnostic”, “collaborative perception” and “scene completion”. We explained and clarified our ideas with references to the prior works in this field.
>
> 4. We also find the VQ-VAE, as the reviewer MFnL suggested, to be a very effective realization of **our proposed collaborative completion framework**. We have implemented it and updated the quantitative results accordingly.
>
> 5. We changed our title to "Multi-Robot Scene Completion: Towards Task-Agnostic Collaborative Perception" to make it more concise.